# Fabrication and Characterization of Transparent and Scratch-Proof Yttrium/Sialon Thin Films

**DOI:** 10.3390/nano10112283

**Published:** 2020-11-18

**Authors:** Amar Kamal Mohamedkhair, Abbas Saeed Hakeem, Qasem Ahmed Drmosh, Abdul Samad Mohammed, Mirza Murtuza Ali Baig, Anwar Ul-Hamid, Mohammed Ashraf Gondal, Zain Hassan Yamani

**Affiliations:** 1Physics Department, King Fahd University of Petroleum & Minerals, Dhahran 31261, Saudi Arabia; amar.salih@kfupm.edu.sa (A.K.M.); magondal@kfupm.edu.sa (M.A.G.); zhyamani@kfupm.edu.sa (Z.H.Y.); 2Center of Excellence in Nanotechnology, King Fahd University of Petroleum & Minerals, Dhahran 31261, Saudi Arabia; ashakeem@kfupm.edu.sa; 3Mechanical Engineering Department, King Fahd University of Petroleum & Minerals, Dhahran 31261, Saudi Arabia; samad@kfupm.edu.sa (A.S.M.); mmurtuza@kfupm.edu.sa (M.M.A.B.); 4Center for Engineering Research, Research Institute, King Fahd University of Petroleum & Minerals, Dhahran 31261, Saudi Arabia; anwar@kfupm.edu.sa

**Keywords:** yttrium, sialon, thin films, ultra-hard, pulsed laser deposition

## Abstract

Transparent and amorphous yttrium (Y)/Sialon thin films were successfully fabricated using pulsed laser deposition (PLD). The thin films were fabricated in three steps. First, Y/Sialon target was synthesized using spark plasma sintering technique at 1500 °C in an inert atmosphere. Second, the surface of the fabricated target was cleaned by grinding and polishing to remove any contamination, such as graphite and characterized. Finally, thin films were grown using PLD in an inert atmosphere at various substrate temperatures (RT to 500 °C). While the X-ray diffractometer (XRD) analysis revealed that the Y/Sialon target has β phase, the XRD of the fabricated films showed no diffraction peaks and thus confirming the amorphous nature of fabricated thin films. XRD analysis displayed that the fabricated thin films were amorphous while the transparency, measured by UV-vis spectroscopy, of the films, decreased with increasing substrate temperature, which was attributed to a change in film thickness with deposition temperature. X-ray photoelectron spectroscopy (XPS) results suggested that the synthesized Y/Sialon thin films are nearly homogenous and contained all target’s elements. A scratch test revealed that both 300 and 500 °C coatings possess the tough and robust nature of the film, which can resist much harsh loads and shocks. These results pave the way to fabricate different Sialon doped materials for numerous applications.

## 1. Introduction

Transparent screens and covers are essential components of various technology products such as solar cells, liquid crystals displays, electrochromic windows, and touch screens and many other applications [1,2,3,4]. These screens and covers are crucial for protecting sensitive components of these devices while still allowing light to pass. However, these screens and covers are often made from materials that are brittle, have poor resistance to scratches and fingerprints, and are prone to optical dimming [5]. Failure of these screens and covers, primarily through mechanical means such as cracking, scratching, shattering, warping, and the like can severely damage or destroy the entire device. Currently, an estimated over 145 million smart-phone screens break every year (2 screens per second in the USA alone). Cracked and scratched screens account for 56% of reported damages, and faulty durability accounts for 30% of the end user’s “top-notch frustrations”. Presently, there is an increased interest in methods to improve the surface properties of transparent materials (glass/polymers) to make consumer products more functional and durable. Mostly, conventional transparent surfaces do not meet all their application requirements. To improve the properties and performance of traditional transparent materials, one approach is to modify the surface [6,7,8,9]. Various methods have been adopted to achieve enhanced surface characteristics including ion exchange [10], gas treatment [11], or by means of thin film coating [12,13,14,15], which were well elaborated in the literature.

Ceramic coatings are considered attractive for the above applications since they possess nonpermeable surface, high color stability, high hardness (scratch and abrasion), and adequate resistance to corrosion, sunlight, and environmental and high temperature degradation. In addition, these coatings are aseptic, hygienic, easy to clean and most importantly, environmentally friendly [13,16,17,18,19,20].

Sialon thin films, which are a solid solution of Si, Al, O, and N, have received significant attention due to their unique mechanical properties including high wear resistance, low coefficient of friction, and high decomposition temperature [21,22]. Sialon thin films are already employed in the industry for various anti-reflection purposes and scratch-protection applications, owing to their reflective coatings’ mechanical strength, thermal resistance, and chemical inertness [21,23,24,25]. The surface of the Sialon thin films can be modified and imparted with added value by introducing alkaline earth/lanthanide metals into the surface. This can be achieved by depositing a thin film coating of M-Sialon (where M = alkaline earth (AE) e.g., Mg, Ca, Sr, Ba, Y and rare earth (RE), e.g., La, Eu, Yb etc.) on the surface. This protection effectively enhances the mechanical and chemical durability of material surfaces [26,27,28]. Sharafat et al. [29] prepared transparent (in the visible region), high refractive index (RI: 2 at 633 nm wavelength) and high hardness (21 GPa) MgSiON thin films using reactive co-sputtering of Si and Mg targets.

Recently, Merkx et al. [23] fabricated transparent Sialon:Eu thin films using three sputtering sources (Al, Eu, Si) in a reactive Ar + N_2_ + O_2_ at room temperature followed by rapid heat treatment to improve the optical properties of Sialon thin films. The sputtering technique, however, suffers from certain disadvantages such as a low deposition rate of the film and challenges in controlling the stoichiometry of the deposited material. Commonly, the main issues associated with the transparent surface are brittleness, weak resistance to scratching, fingerprints, and optical dimming. Pulsed laser deposition (PLD) technique has been widely used for deposition of metal oxides, metal nitrates etc. The most important advantage of this method is the ability to transfer the material stoichiometry from a multi-component ablation target to a growing film because ablation occurs as soon as the target is irradiated with a laser beam. Furthermore, the films fabricated by PLD technique after the plasma transfer are usually very adherent to substrate along with a rather large hardness. Compared to most physical vapor deposition (PVD) techniques, PLD can be used easily to fabricate ultra-thin multi-layer films. The deposition of the coating of AE/RE-Sialon ceramic by PLD and further understanding of this material is suggested in order to overcome these surface-related issues.

In this work, we developed a novel method to prepare transparent yttrium containing Sialon thin films using pulsed laser deposition for high technology applications. The morphological, compositional, structural, and mechanical properties of the fabricated thin films were studied as a function of substrate temperature (room temperature (RT) to 500 °C).

## 2. Materials and Methods

### 2.1. Fabrication of Y/Sialon Target by Spark Plasma Sintering (SPS)

A mixture of starting materials was reacted together for the preparation of Sialon samples. Sialon matrix consisted of commercially available powders (Table 1) namely; Y_2_O_3_ (<300 nm, 98%, Sigma Aldrich, St. Louis, MO, USA), SiO_2_ (20–50 nm, 99.5%, Sigma Aldrich, St. Louis, MO, USA), AlN (<100 nm, Sigma Aldrich, St. Louis, MO, USA), and Si_3_N_4_ (300–500 nm, 95% α-phase content, UBE Industries SN-10, Tokyo, Japan). Table 1 summarizes the weight and molar percentages (wt.%) of the powder precursors. A composition is based on the general formula, Y*_m_*_/3_Si_12-(*m*+*n*)_Al_(*m*+*n*)_O_(*n*)_N_16−*n*_ and with *m* = 1.0, and *n* = 1.2. Probe sonicator (Model VC 750, Sonics, Newtown, CT, USA) was employed to achieve uniform mixing of the powders in ethanol. After 20 min of probe sonication, the powder mixture was oven-dried at 90 °C temperature for about 20 h to evaporate the ethanol. After drying, the mixtures were consolidated using spark plasma sintering (SPS) equipment (FCT system, model HP D5, Rauenstein, Germany). The powders were synthesized in a 20 mm graphite die under a uniaxial pressure of 50 MPa. The synthesis temperature, heating rate, and soaking time at which all the samples were sintered are 1500 °C, 100 °C/min and 30 min, respectively.

After the SPS processing, the graphite contamination on the surface of the synthesized samples was removed using SiC abrasives (grit sizes ranging from 180 grit to 1000 grit). Furthermore, the samples were ground and polished using a diamond grinding wheel to prepare a scratch-free surface. To obtain a mirror-like surface, the scratch-free samples were polished using alumina suspension (particle size 0.3 µm) on a polishing cloth.

The established Archimedean principle of density measurement was employed to determine the density of the synthesized sample. A Mettler Toledo kit was used to determine the density using distilled water as the media. The average of five results is presented in the range of 3.31 g/cm^3^. Universal hardness tester developed by Zwick-Roell (ZHU250 Germany) was used to determine the hardness of the synthesized sample (later used as a PLD target) at a load of 10 kg and resulting hardness was Hv10 16 GPa. Using the maximum crack length (*d*) and the hardness value obtained, the indentation fracture toughness (*K_IC_*) was obtained using the Evans criterion stated below in Equation (1), where ‘MCL’ represents the maximum crack length and resulting *K_IC_* is 6.87 MPa·m^½^.
(1)$$K_{1C}=0.48{\left(\frac{MCL}{d_{1/2}}\right)}^{-1.5}\;\frac{HV_{10}\;\sqrt{d/2}}{3}$$

Thermal expansion equipment (Mettler Toledo, TMA/SDTA-LF/1100, Switzerland) was used to compute the coefficient of thermal expansion of the synthesized sample. The sample used for this experiment was cut to finished dimensions of 4 mm× 4 mm × 4 mm each and the value measured was 3.24 ppm·K^−1^.

### 2.2. Fabrication of Y/Sialon Thin Films

Y/Sialon thin films were fabricated with pulsed laser deposition (PLD, Neocera, Beltsville, MD, USA) using KrF laser operating at a wavelength (*λ*) = 248 nm, energy 200 mJ, laser fluence = 0.1 J/cm^−2^, and frequency = 10 Hz. The distance between the substrate and the target was approximately 6 cm. The substrate temperature was changed from RT up to 500 °C to study the effect of the deposition temperature on the physical and mechanical properties of Sialon thin films. Before deposition, the substrates were ultrasonically cleaned in acetone, and ethanol, respectively. Y/Sialon thin films were grown on soda-lime silicate glass and silicon wafer with a dimension of 20 mm × 20 mm. The base pressure was 0.1 × 10^−6^ Torr, while the deposition was carried out under an argon atmosphere with 1.3 × 10^−3^ Torr working pressure for 2 h.

### 2.3. Characterization Techniques

Sialon target and thin film crystal structure were examined with X-ray diffractometer system (XRD, Rigaku Miniflex 600, Tokyo, Japan) with Cu Kα radiation *λ* = 1.5406 Å. The experiment was carried out in a 2*θ* range of 10 to 90° with a scan speed of 0.2°/min. The morphological properties of the target and the fabricated thin film samples were explored using field emission scanning electron microscope (FESEM) (Tescan Lyra 3, Brno, Czech Republic) and atomic force microscopy (AFM) (Nanosurf Easyscan, Liestal, Switzerland). The microstructure of the films was investigated by a field emission transmission electron microscope (FETEM) (JEOL-JEM2100F, Tokyo, Japan) with an accelerating voltage of 200 kV. The chemical composition of the thin films deposited on silica wafers substrate was investigated with the X-ray photoelectron spectroscopy (XPS) technique, (Thermo Fisher Scientific, model: ESCALAB250Xi, Waltham, MA, USA). Optical transmittance was carried out by employing UV/Vis spectrophotometer (Jasco V-570, Tokyo, Japan) in the wavelength range of 300–1200 nm. The scratch test was performed using a ramping/progressive load whereby the load is increased linearly from 0 to a maximum load of 15 N with a loading rate of 15 N/min. The indenter was a standard Rockwell C indenter with a 100 μm tip radius. The scratch length was set to 5 mm with a traversing speed of 5 mm/min. The instrument records and displays the acoustic emission signal (AE) graph, along with the variation of coefficient of friction and frictional force graphs along with the scratch, which are used in conjunction with the micrograph of the track to determine the lower (*L*_c1_) and the upper (*L*_c2_) critical loads. The lower critical load (*L*_c1_) is defined as the load at which the crack within the coating is initiated, which corresponds to the cohesive failure in the coatings and the upper critical load (*L*_c2_) is defined as the load at which the coating is completely delaminated from the substrate corresponding to the adhesive failure at the interface between the coating and the substrate. In the present study, *L*_c2_ is taken as the maximum load that the coating can sustain before complete failure or delamination.

## 3. Results and Discussion

### 3.1. Microstructure and Phase Analysis

Figure 1 displays the XRD spectrum obtained from the Y/Sialon target and Y/Sialon thin films fabricated using PLD technique. As can be observed in Figure 1, the Y/Sialon target showed well-defined diffraction peaks that indicated the formation of β-Sialon phase. Peaks were observed at 2θ = 13.70°, 23.60°, 27.28°, 28.79°, 33.77°, 36.23°, 39.05°, 41.50°, 47.61°, 47.92°, 49.97°, 52.22°, 56.43°, 57.90°, and 59.74° matched well with the standard JCPDS (48–1615) [25]. Freshly fabricated Y/Sialon thin films deposited at RT, 100 °C, 300 °C, and 500 °C showed no diffraction peaks and thus confirming the amorphous nature of fabricated thin films.

Figure 2 shows the influence of the substrate temperature on the morphology of the Y/Sialon thin films fabricated using PLD. The Y/Sialon film deposited at RT (25 °C) (Figure 2a) possessed a thickness of approximately 18 nm and grain size of 20 nm. It appeared consistently uniform and smooth while exhibiting accumulation of spherical nanoparticles of two distinct dimensions at the surface. Similar features were observed in the case of Y/Sialon thin films deposited at 100 °C (Figure 2b) with the exception that the thickness of the synthesized film was relatively higher i.e., approximately ~35 nm. A further increase in the substrate temperature of the film to 300 °C increased the number of the small-sized accumulated nanoparticles at the surface (Figure 2c). Besides, the thickness of the sample was sharply increased to reach approximately ~135 nm. At the substrate temperature of 500 °C (Figure 2d), not only the obtained film became more uniform and smoother, the dimensions of large-sized accumulated spherical nanoparticles increased with concurrent disappearance of the smaller ones.

The surface morphologies in 5 µ × 5 µ scanning area of the Y/Sialon thin films as a function of substrate temperature used during deposition are investigated via AFM and displayed in Figure 3a–d. The root mean square (RMS) roughness of the thin films are summarized in Table 2. It is found that the RMS roughness of Y/Sialon thin films decreases with increasing substrate annealing temperature. Generally, a decrease in RMS roughness leads to good homogeneity in thin films [29]. Such observation could be attributed to the diffusion of small accumulated nanoparticles at the surface of smooth thin film and into fewer larger particles as confirmed by the FESEM analysis.

Transmission electron microscope (TEM) was employed to examine the structure of the films prepared at 100 and 500 °C. Selected films were studied to evaluate their structural integrity and to detect the presence of any inhomogeneity in the film structure. TEM images in Figure 4 show the microstructure of the film, prepared at 100 °C, at various magnifications. There was no inhomogeneity observed except for slight variation in the thickness of the film. Additionally, there was no evidence for the presence of any ordered lattice fringes or any other substructural features in the film (Figure 4c). Selected area electron diffraction (SAED) patterns confirmed the amorphous nature of the film (Figure 4d) which corroborates the results obtained from XRD analysis.

Figure 5 shows the microstructure of samples annealed at 500 °C at various magnifications. Most of the film exhibits a uniform surface and an amorphous. However, HRTEM images in Figure 5c,d) show the existence of a few nanocrystallites with ordered lattice fringes within the amorphous matrix of the film. The presence of spots in the electron diffraction pattern shown in Figure 5e confirms the formation of nanocrystallites within the amorphous film. Furthermore, ordered lattice fringes reveal an atomic d-spacing of ~0.2514 nm, which provides evidence for the presence of beta-sialon nanocrystallites (2*θ* = 35.90°) and d-spacing of 0.249 nm (210). Likewise, XPS analysis confirmed the higher atomic ratio of nitrides present in the structure. The size of nanocrystallites ~5 nm, along with the film thickness, probably interferes and influences the transmittance and related properties of the films.

Figure 6 shows the optical transmittance spectra obtained from Y/Sialon films prepared with PLD in the wavelength range (300–1200 nm). In general, high transmittance of thin films is an indication of good homogeneity as well as low surface roughness [30]. It is clear that the spectra of the as-deposited Y/Sialon films showed excellent transmittance in the UV, visible, and NIR regions. The average optical transmittance of the as-deposited Y/Sialon films in the UV (300–400 nm), visible (401, 750 nm), and NIR (751, 1200 nm) were 91.6%, 92.7%, and 95.5%, respectively. As the substrate temperature of the Y/Sialon films increased to 100 °C, a slight decrease in transmittance was observed. For example, the average optical transmittance in the visible region was 91.4%, which is merely 1% less than for the coating deposited at RT. For the Y/Sialon films deposited at 300 and 500 °C, the optical transmittance from the NIR to 600 nm was significantly decreased compared with other films. The reduction in the transmittance could be attributed to the increase in the grain size, and the thickness of the films as confirmed by FESEM cross-section and AFM images. Interestingly, the optical transmittance of the Y/Sialon films deposited at 300 and 500 °C declined dramatically below 600 nm, which might be attributed to the shift of the Y/Sialon bandgap to the UV region. Indeed, Boyko et al. [31] reported that the direct bandgaps of the β-Si_6−*z*_ Al*_z_*O*_z_*N_8−*z*_ (*z* = 0.0, 2.0, and 4.0) fabricated using ultra-hot isostatic pressing were about 7.2, 6.2, and 5.0 eV and the bandgap reduction could be attributed to the movement of the *Op*-states toward the Fermi level.

The chemical composition of the Y/Sialon thin films was obtained with X-ray photoelectron spectroscopy (XPS). Each XPS spectrum was corrected for steady-state charging by aligning C1s to 284.60 eV. Typical XPS survey spectra of the fabricated films and O1s, Si2p, Al2p, N1s, and Y3d core-level spectra for the Y/Sialon thin film prepared using PLD at 500 °C are displayed in Figure 7. As can be observed in the XPS survey spectra shown in Figure 7a, all the constituent elements (Si, Al, N, O, and Y) were observed. Argon, which could be trapped in the interstitial sites during the growth process or during the XPS etching was detected in all samples. The binding energy, full width at half maximum (FWHM), area and the weight of each component in the samples are listed in Table 3. It can be noticed from the table that there are no significant binding energy shifts. Furthermore, the Si:Al and the O:N atomic ratio in the Y/Sialon films are nearly equal. These observations suggest that the synthesized Y/Sialon thin films are nearly homogenous and form a complete solid solution. Figure 7b–f shows the high resolution XPS spectra obtained from Y/Sialon thin film prepared using PLD system at 500 °C deposition temperature. The Si2p XPS spectrum (Figure 7b) decomposed into two components attributed to a comprehensive contribution of Si–N bonds, and a little of Si–O bonds. The XPS spectrum of Al2p (Figure 7c) showed the presence of two peaks centered at 74.49 eV and 75.00 eV with similar atomic% corresponding to Al–N and Al–O bonds, respectively. The XPS spectrum for O1s revealed the presence of one peak corresponding to Si-O bonds (Figure 7d). Besides, the deconvolution of the XPS N1s peak (Figure 7e) further confirmed the presence of N-Si and little N-Si-O bonds. The deconvolution of the XPS spectra in the Y3d region (157–165 eV) shown in Figure 7f displays two peaks located at 158.67 eV, and 160.69 eV that correspond to Y3d_5/2_ and Y3d_3/2_ of Y_2_O_3_ nanoparticles, respectively. The peak area ratio for the Y3d_5/2_ and Y3d_3/2_ is about 1.43, which is very close to the theoretically expected value of 1.5 based on the degeneracy of the states. 

### 3.2. Effect of Substrate Temperature on the Scratch Resistance of Coatings

Synthesized coatings were subjected to a scratch test in order to evaluate their adhesive strength, which is a measure of their scratch resistance. Figure 8 shows the scratch data plot for the coating developed at room temperature, which exhibits applied ramping normal load, along with the acoustic signal that accompanies the scratch. The panoramic view (Figure 8b) of the scratch and the zoomed-in views (Figure 8c,d) of the scratch at the location of L*c*_1_ (failure/crack initiation) and L*c*_2_ (failure/delamination) [32].

The analysis of the scratch was carried out by taking into consideration all the elements of the data such as the acoustic signal and the microscopic images of the scratch. The onset of cracking or the initial damage can be seen in Figure 8c, which is the zoomed-in view of the scratch, corresponding to a load of 8.1 N, which is assigned as the first critical load (*Lc*_1_). This primary damage has the shape of interfacial shell-shaped spallation. It is also to be noted that *Lc*_1_ corresponds to the first small jump on the acoustic emission signal [33]. The second critical load (*Lc*_2_) of 13.8 N is found along with the scratch. Figure 8d at a point at which the damage becomes continuous such as hertz cracking where microcracks are observed within the scratch groove resulting in the delamination of the coating and its complete failure. Moreover, after this point, all the acoustic emission signal becomes noisier. A similar analysis was conducted on all the coatings to determine both the critical loads (*Lc*_1_ and *Lc*_2_) and to get the maximum load (*Lc*_2_) that the coatings sustain before failure. Figure 9 shows the zoomed-in images of the other coatings deposited at 300 and 500 °C, respectively, corresponding to the positions of the critical loads.

It was observed that the scratch resistance of the coating deposited at 300 °C did not significantly change as compared to the coating deposited at room temperature (RT). However, as the deposition temperature increased to 500 °C, delamination, or complete failure of the coating along the scratch was not observed whereby *Lc*_2_ location could not be determined, suggesting a significant improvement in the scratch resistance of the coating.

In summary, the current study, presented a few results on ablation and deposition of Y/Sialon onto glass substrates. It was observed that the surface condition and processing temperature plays a vital role in the adhesion and other properties of the deposited films at various temperatures. It was observed that PLD at higher deposition temperatures tends to enhance both thickness and adhesion properties of the films. Substrate heating during processing modified the amorphous nature to nanocrystallites, resulting in improved adhesion and surface properties. Table 4 summarizes the properties of similar film materials and deposition processes from the literature, which help in viewing the better performance of the current results as compared to the existing data.

## 4. Conclusions

In this work, transparent and scratch-proof Y/Sialon thin films were deposited at various substrate temperatures using PLD of Y/Sialon target which was fabricated by spark plasma sintering. The deposited films were studied by using FESEM, TEM, XRD, XPS, AFM, and UV-vis spectroscopy techniques. The scratch test was investigated using a ramping/progressive load whereby the load is increased linearly from 0 to a maximum load of 15 N with a loading rate of 15 N/min. The UV-vis results demonstrated that the transparency of the films was high and decreased with increasing substrate temperature, which could be attributed to a change in film thickness with deposition temperature. FESEM and AFM images illustrated that the coating is homogenous, continuous over a large area without voids. Y/Sialon target exhibited a higher hardness (HV_10_) 16 GPa and fracture toughness was (*K*_IC_) 6.87 MPa·m^1/2^. A coefficient of thermal expansion 3.24 ppm·K^−1^ which is little less than silicon nitride (Si_3_N_4_) and density of 3.31 g/cm^3^ little high than Si_3_N_4_. A scratch test revealed that both 300 and 500 °C coatings possess the tough and robust nature of the film, which can resist much harsh loads and shocks. According to the present studies, it was found that fabricated films have the potential for use in high technology applications, including protector for smart-phone screens, army vehicle windows, and liquid crystal displays.

## Figures and Tables

**Figure 1 nanomaterials-10-02283-f001:**
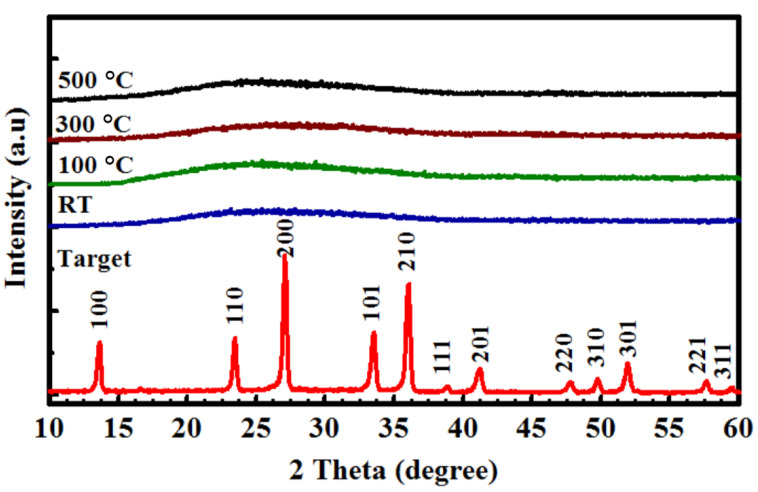
X-ray diffractometer (XRD) spectra obtained from Y/Sialon target prepared using spark plasma sintering (SPS) and Y/Sialon thin films fabricated using pulsed laser deposition (PLD).

**Figure 2 nanomaterials-10-02283-f002:**
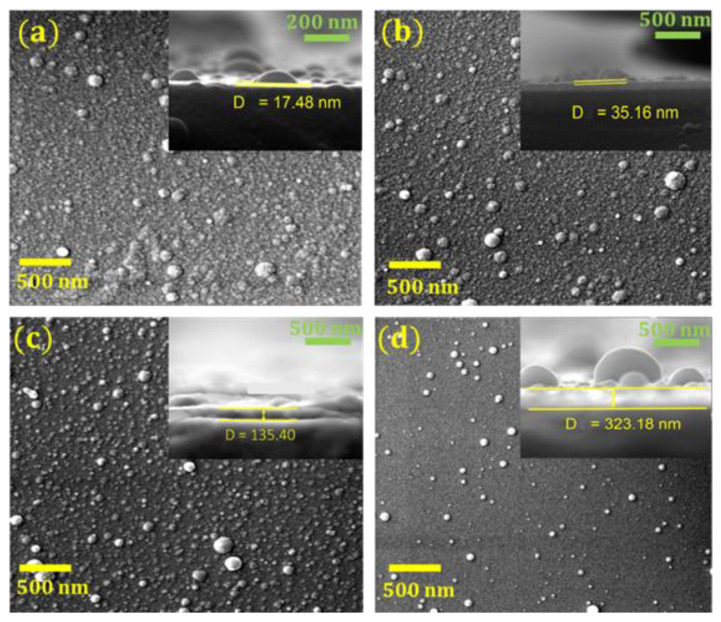
Field emission scanning electron microscope (FESEM) images obtained from the Y/Sialon thin films prepared with PLD at (**a**) room temperature (RT), (**b**) 100 °C, (**c**) 300 °C, and (**d**) 500 °C. Insets correspond to cross-sectional images.

**Figure 3 nanomaterials-10-02283-f003:**
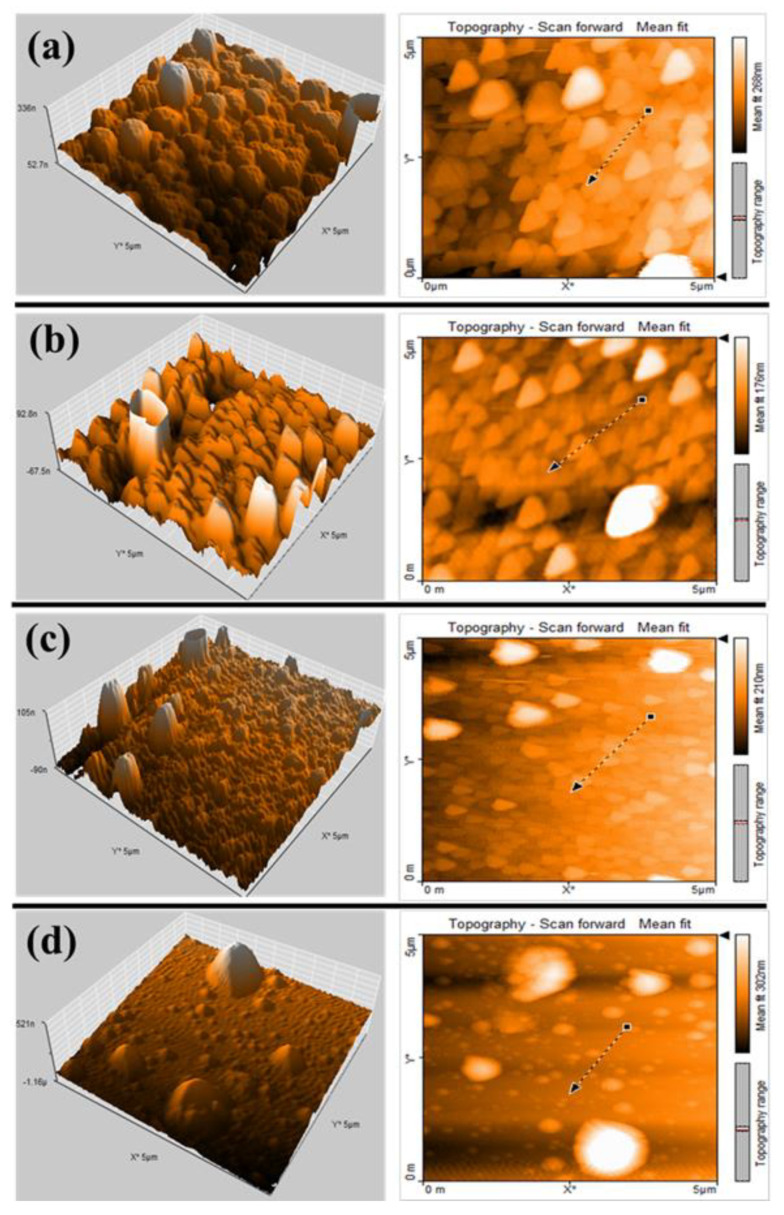
2D and 3D atomic force microscopy (AFM) images of the Y/Sialon thin films prepared by PLD at: (**a**) RT; (**b**) 100 °C; (**c**) 300 °C; (**d**) 500 °C.

**Figure 4 nanomaterials-10-02283-f004:**
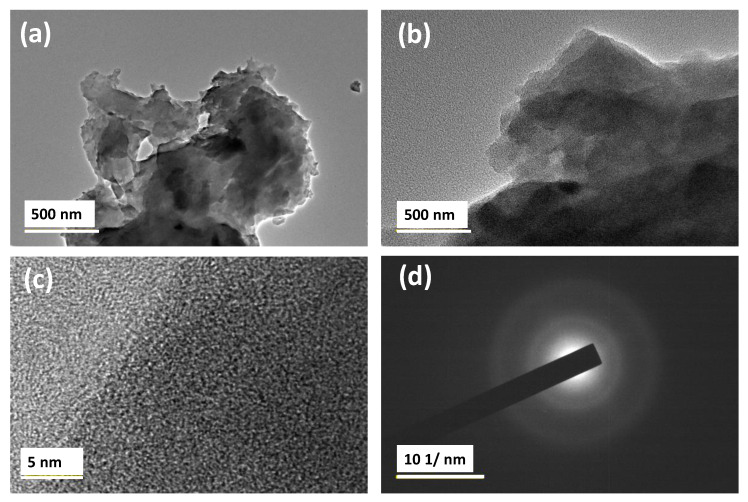
Y/Sialon thin films annealed at 100 °C TEM images at various magnifications (**a**–**c**) along with the corresponding selected area diffraction pattern from image (**d**).

**Figure 5 nanomaterials-10-02283-f005:**
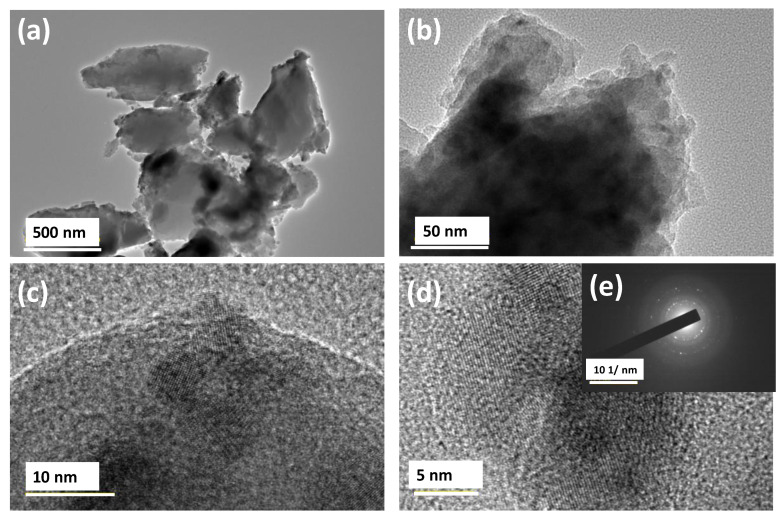
PLD at 500 °C transmission electron microscope images at various magnification (**a**–**d**) along with the corresponding selected area diffraction pattern (insert) from image (**e**).

**Figure 6 nanomaterials-10-02283-f006:**
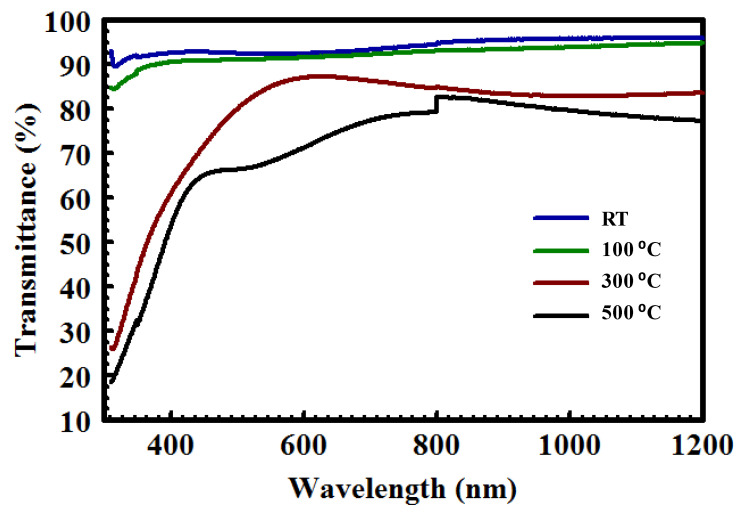
UV-Vis transmittance spectra obtained from Y/Sialon thin films prepared using PLD at RT, 100 °C, 300 °C, and 500 °C.

**Figure 7 nanomaterials-10-02283-f007:**
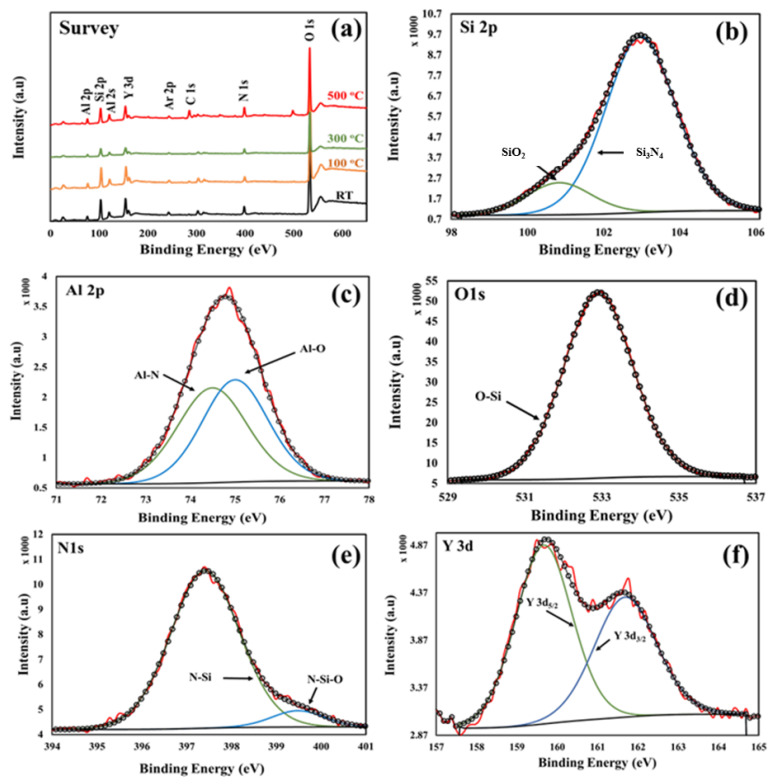
(**a**) XPS survey of the Y/Sialon thin films prepared using PLD at various deposition temperatures; (**b**–**f**) high resolution scan of Si2p, A12p, O1s, N1s, and Y3d, respectively.

**Figure 8 nanomaterials-10-02283-f008:**
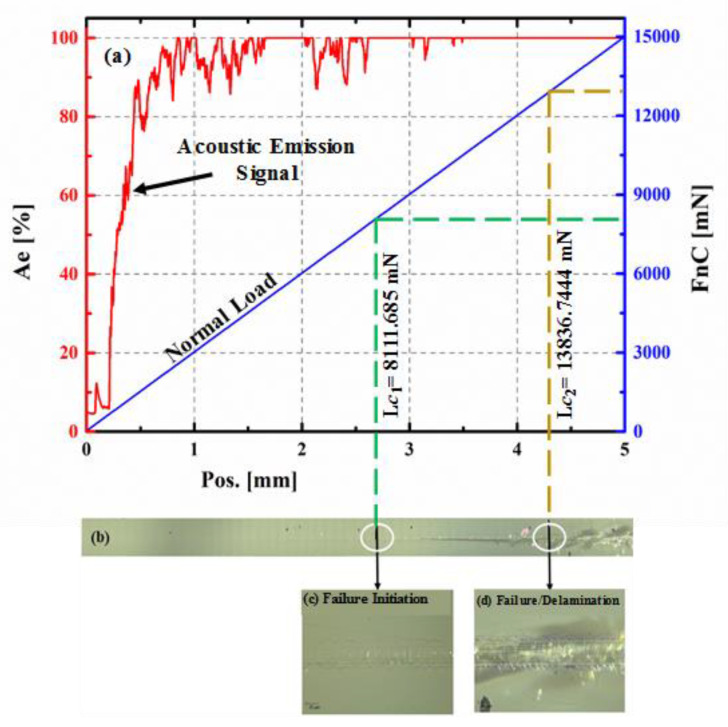
(**a**) A plot showing applied ramping normal load and the acoustic signal generated during the scratch test; (**b**) panoramic view of the scratch; (**c**) zoomed-in view of the scratch at a location of *Lc*_1_ (failure/crack initiation); (**d**) zoomed-in view of the scratch at location *Lc*_2_ (failure/delamination); for the coating deposited at room temperature.

**Figure 9 nanomaterials-10-02283-f009:**
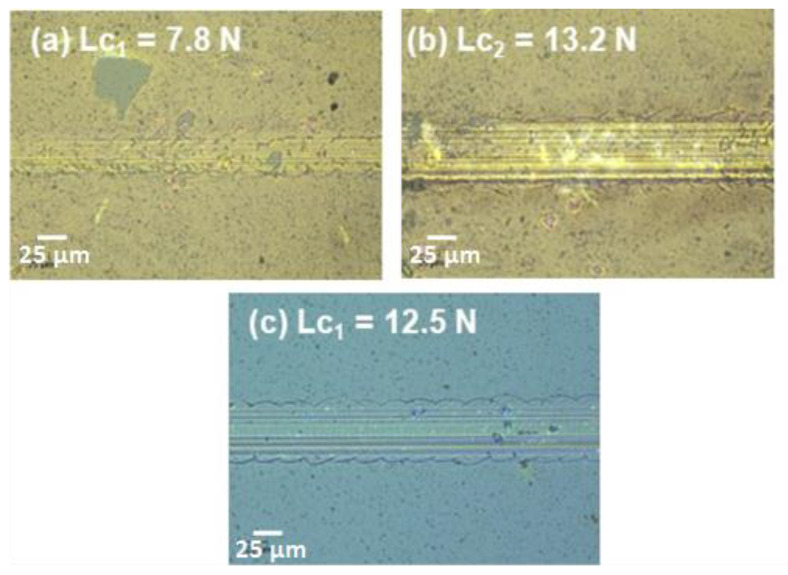
(**a**,**b**) *Lc*_1_ and *Lc*_2_ location along with the scratch for the coating deposited at 300 °C; (**c**) The *Lc*_1_ location along with the scratch for the coating deposited at 500 °C.

**Table 1 nanomaterials-10-02283-t001:** Weight (in gram) and molar percentage of precursors to synthesize 6 g of sample (composition: *m* = 1.0, *n* = 1.6) XX-1016 Series, Y*_m_*_/3_Si_12-(*m*+*n*)_ Al_(*m*+*n*)_O_(*n*)_ N_16−*n*_ = Y_0.333_Si_9.4_Al_2.6_O_1.6_N_14.4_.

Chemicals	α-Si_3_N_4_	AlN	Y_2_O_3_	SiO_2_	Al_2_O_3_
**Wt. (g)**	4.3550	0.9003	0.3822	0.1342	0.2282
**WT. (%)**	72.58	15.01	6.37	2.24	3.80
**Elements**	Si	Al	Y	O	N
**Mole%**	33.1769	9.1766	1.1753	5.6471	50.8241

**Table 2 nanomaterials-10-02283-t002:** Physical properties of Y/Sialon thin films prepared using PLD technique at various deposition temperatures.

	Sample	Thickness (nm)	RMS (nm)
1	Y- Sialon (RT)	18	16.0
2	Y- Sialon (100 °C)	35	14.5
3	Y- Sialon (300 °C)	135	11.4
4	Y- Sialon (500 °C)	323	9.7

RMS—root mean square; RT—room temperature

**Table 3 nanomaterials-10-02283-t003:** XPS analysis of the Y/Sialon thin films prepared using PLD technique at evaluated substrate temperatures.

	Name	Peak BE	FWHM eV	Area (P) CPS.eV	Atomic %
**Y/Sialon** **RT** °C	O1s	532.91	3.17	1,040,178	48.9
	Si2p	103.25	3.24	210,528.4	30.84
	N1s	398.3	3.23	87,008.34	6.32
	Al2p	75.86	3.02	38,248.16	8.46
	Y3d	154.25	3.57	270,047.6	5.48
**Y/Sialon** **100** °C	O1s	532.84	3.22	1,035,674	48.79
	Si2p	103.14	3.26	207,127	30.4
	N1s	398.28	3.41	64,431.87	4.69
	Al2p	75.85	3.02	48,034.71	10.65
	Y3d5	154.14	3.64	269,156.5	5.47
**Y/Sialon** **300** °C	O1s	533.16	3.06	384,019.3	43.71
	Si2p	103.25	3.6	90,340.59	32.04
	N1s	398.87	2.97	47,364.59	8.33
	Al2p	76.2	2.87	19,389.59	10.39
	Y3d	154.18	4.14	112,483.3	5.52
**Y/Sialon** **500** °C	O1s	532.99	3.13	644,686.3	40.88
	Si2p	102.99	3.41	158,230.6	31.26
	N1s	398.77	3.09	94,786.28	9.29
	Al2p	75.99	2.99	45,746.21	13.65
	Y3d	153.97	3.75	179,897.7	4.92

**Table 4 nanomaterials-10-02283-t004:** The mechanical properties (scratch resistance, hardness, and Fracture Toughness) of thin film materials reported in the literature.

#	Substrate	Coating	Technique	Film Thickness (nm)	Scratch Resistance	Hardness (GPa)	Fracture Toughness	Comment	Ref.
1	Float glass	MgSiON	Co-sputtering	372–463	Not reported	Coating = 21 Substrate = 7	not reported	E(coating) = 166 GPA E(substrate) = 72 GPA	[5]
2	Uncoated or Pt-coated sapphire	SiAlON	RF magnetron sputtering	200	not reported	not reported	not reported	films are very wear resistant than sapphire	[21]
3	Zinc sulfide window materials	SiAlON	Ion beam sputtering method.	633	not reported	film = 7.1 Subs = 4.1	not reported	micro hardness of the zinc sulfide was improved by 75% on average after being coated with SiAlON films	[34]
4	Glassy carbon substrates	None	DC magnetron sputtering	200	not reported	subs = 3 Film = 10	not reported		[35]
5	HK40 (Fe-Ni-Cr) Alloy	Silicon Carbide	Pulsed Laser Deposition	1000	InitialFailure @ 0.48 N Complete delamination @ 4.37 N	not reported	not reported	Shot peening and heating of surface of substrate improves both coverage and adhesion of film	[36]
6	p-type Silicon Wafers	Diamond like Carbon Films Copper Doped)	Pulsed Laser Deposition	400–600	Improved adhesion compared to undoped film	38	not reported	A constant load of 10 g was used for scratch test.	[37]
7	Silicon Wafers	Carbon Nitride	Plasma Assisted Pulsed Laser Deposited	NA	Initial Failure @ 80 mN	13.5	not reported	E = 100 GPa	[38]
8	Silicon &NaCl	silicon oxynitride	Pulsed Laser Deposition	NA	not reported	not reported	not reported		[39]
9	NA	Eu Doped SiAlON	Pulsed Laser Deposition	150	Not reported	Not reported	Not reported		[40]
10	Si & Pt-coated Si substrates.	SiO_2_ and SiO_x_N_y_	Reactive Pulsed Laser Deposition	10–2000	not reported	not reported	not reported		[41]

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
