# Peer review of "Fabrication and Characterization of Transparent and Scratch-Proof Yttrium/Sialon Thin Films"

_nanomaterials, 2020, doi:10.3390/nano10112283_

Round 1

Reviewer 1 Report

The manuscript submitted by Mohamedkhai et al. presents a work on the preparation and characterization of films made of Y and SiALON obtained by PLD. Overall the paper is well structure, written and the conclusions are adequately supportted on the experiments. This reviewer would recommend publication of their manuscript after just some minor revisions addressed to the authors as follows:

  1. Please correct the term "gm" for grams when appropriate for "g". There are several place where it occurs.
  2. Correct the sub and super-indices where necessary.
  3. In table I, weight and content of precursor materials used for the preparation of the films, please indicate what was the final mass of material prepared for which the weights given were employed. I would also recommend to replace the wt. % of each component by the molar % or the atomic % of the elements instead.
  4. Did the authors conclude on any variation of the N incorporated into the films after the preparation?
  5. Could the authors indicate the heating rate of the measurement of coefficient of expansion?
  6. In line 173, page 4, the authors mentioned that the thickness of the later deposited at 100ºC was slightly higher than the one obtained at room temperature. However, it is to note that the value at 100ºC represents the double of that at RT, not only slightly higher. Could they have a proper explanation for that? Is ti possible that they provide a representation of the variation in thickness with the deposition temperature? Does it follow a particular law through which it is possible to predict and control the thickness of the films?
  7. In line 207, page 7, the authors used "amorphous structure". Please remove structure from the sentence. The material will be amorphous, which means it does not have a periodically ordered strucutre, but the structure itself is not amorphous.
  8. In line 225, page 8, please remove "there was".
  9. In line 232, page 8, remove "due to".
  10. In line 248, page 9, remove "are observed".
  11. In the last sentences of page 9, the authors stated that the synthezised films are nearly homogeneous and form a complete solid solution. Thereafter, they also present that the XPS spectra would be compose of signals as due to the presence of the elements under different chemical compounds, such as SiO2, Si3N4 or AlN. This last view of the spectral analysis confronts with the view of a "complete solid solution" as they stated in the lines before. Could they explain why the film must be assumed as an homogeneous solid formed by a solid solution but where the signals of each of the forming elements can be represented under environments assigned to particular compounds with much different chemical composition?

Reviewer 2 Report

Manuscript is devoted to production and characterisation of Y/Sialon thin films. The film formation was by pulsed laser deposition (PLD), technique that enables formation of thin films with composition close to target material but does not likely to be suitable for large area deposition. However, the material and way of deposition are very interesting for many technological applications. The experimental techniques using for characterisation are appropriate and well described, manuscript is correctly organized.

However, before publishing I have some suggestions that are listed below.   

The comparison between results obtained here with other published results regarding  scratch resistance of coatings and regarding deposition rate by actual PLD technique and some others  would improve the quality of the manuscript.

Conclusion and Abstract look almost the same that should be corrected.

Type mistakes must be corrected

  2.3 Characterization techniques

“…or delamination.3.

155 Results

156 3. Results and discussion

157 3.1 Microstructure and...”

  1. Conclusions

”...much harch loads…”

Ref 28 looks strange

Reviewer 3 Report

In the manuscript „ Fabrication and Characterization of Transparent and Scratch-Proof Yttrium/Sialon Thin Films” the authors realized by PLD yttrium/sialon layers. The results are interesting and well presented. However, some improvements can be added:

-Detail the advantages of using PLD technique!

-2.2 Fabrication of Y/Sialon thin films

Sialon thin films were grown on soda-lime silicate glass....

Is a Sialon only film or Y/Sialon film?

-Which was the laser fluence?

- The reduction in the transmittance could be attributed to the increase in the grain size, the number of grain boundaries, and the thickness of the films ...

In my opinion you can not sustain in the same time that the transmittance decreases due to increase in the grain size and the number of grain boundaries. An increase in the grain size attracts a decrease in the number of the grain boundaries.

-I think you can insert the annealing temperature in the SEM, TEM images for a easy  identification of the samples.
